# Exercise-Induced N-Lactoylphenylalanine Predicts Adipose Tissue Loss during Endurance Training in Overweight and Obese Humans

**DOI:** 10.3390/metabo13010015

**Published:** 2022-12-22

**Authors:** Miriam Hoene, Xinjie Zhao, Jürgen Machann, Andreas L. Birkenfeld, Martin Heni, Andreas Peter, Andreas Niess, Anja Moller, Rainer Lehmann, Guowang Xu, Cora Weigert

**Affiliations:** 1Institute for Clinical Chemistry and Pathobiochemistry, Department for Diagnostic Laboratory Medicine, University Hospital Tübingen, 72076 Tübingen, Germany; 2CAS Key Laboratory of Separation Science for Analytical Chemistry, Dalian Institute of Chemical Physics, Chinese Academy of Sciences, Dalian 116023, China; 3Institute for Diabetes Research and Metabolic Diseases of the Helmholtz Zentrum München at the University of Tübingen, 72076 Tübingen, Germany; 4German Center for Diabetes Research (DZD), 85764 Neuherberg, Germany; 5Section on Experimental Radiology, Department of Diagnostic and Interventional Radiology, University Hospital Tübingen, 72076 Tübingen, Germany; 6Department of Internal Medicine IV, University Hospital Tübingen, 72076 Tübingen, Germany; 7Division of Endocrinology and Diabetology, Department of Internal Medicine 1, University Hospital Ulm, 89081 Ulm, Germany; 8Department of Sports Medicine, University Hospital Tübingen, 72076 Tübingen, Germany; 9Interfaculty Research Institute for Sports and Physical Activity, University of Tübingen, 72076 Tübingen, Germany

**Keywords:** N-Lactoylphenylalanine (Lac-Phe), obesity, biomarker, exerkine, exercise intervention

## Abstract

Physical exercise is a powerful measure to prevent cardiometabolic diseases. However, the individual response to lifestyle interventions is variable and cannot, to date, be predicted. N-Lactoylphenylalanine (Lac-Phe) produced during exercise has recently been shown to mediate weight loss in obese mice. Lac-Phe could also contribute to, and potentially explain differences in, the effectiveness of exercise interventions in humans. Sedentary overweight and obese subjects completed an 8-week supervised endurance exercise intervention (n = 22). Before and after the intervention, plasma levels of Lac-Phe were determined by UHPLC-MS in the resting state and immediately after an acute bout of endurance exercise. Adipose tissue volume was quantified using MRI. Acute exercise caused a pronounced increase in Lac-Phe, both before and after the intervention. Higher levels of Lac-Phe after acute exercise were associated with a greater reduction in abdominal subcutaneous and, to a lower degree, visceral adipose tissue during the intervention. Lac-Phe produced during physical activity could contribute to weight loss by acting as a signaling molecule that regulates food intake, as previously shown in mice. Quantification of Lac-Phe during an exercise test could be employed as a tool to predict and potentially improve the individual response to exercise-based lifestyle interventions in overweight humans and those with obesity.

## 1. Introduction

The incidence of obesity and, with it, type 2 diabetes is increasing worldwide. Physical activity and weight loss are two important pillars of diabetes prevention [1]. In addition to increasing energy expenditure and reducing adipose tissue volume, physical exercise exerts a multitude of beneficial effects on the whole-body level and counteracts type 2 diabetes by improving insulin resistance and blood glucose control [2]. There is, however, a large variability in the effectiveness of exercise-based lifestyle interventions regarding therapeutic targets, such as improvement of blood glucose control or reduction of adipose tissue mass, that cannot satisfactorily be explained up to now [2,3,4]. The variability also occurs when exercise is performed in a supervised fashion, ruling out differences in exercise adherence as a confounder, and at an individualized intensity that is typically based on VO_2_peak as a measure of cardiorespiratory fitness. Increasing the “dose” of exercise might be effective in some cases but spare time that can be allocated to exercise is a limiting factor and the studies that have been performed so far allow no general recommendation whether and to what extent volume, intensity or modality, alone or in combination, should be modified to improve, e.g., glucose control [2]. Thus, there is a need to improve the predictability of the success of exercise interventions and for a more personalized approach when designing lifestyle interventions in humans.

The pronounced changes in skeletal muscle and whole-body energy metabolism induced by physical activity cause an increase in the concentration of a multitude of small metabolites in the circulation. A growing number of these metabolites have been found to not only reflect metabolic spillover but also to play an essential role in mediating the crosstalk between different cells and organs and thus, in mediating the beneficial effects of physical activity [5,6]. Thus, after decades in which research aiming to identify novel signal transducers and biomarkers related to exercise adaptation had mainly been focused on proteins, small metabolites are increasingly gaining attention as “exerkines”, i.e., as factors produced during exercise that mediate its beneficial acute and long-term effects in a hormone-like fashion [7].

N-Lactoylphenylalanine (Lac-Phe) is a pseudo-dipeptide generated from lactate and phenylalanine that exhibits a particularly pronounced increase in blood during and shortly after physical exercise [8]. Recently, Lac-Phe has been proposed to be a novel “exerkine” that lowers body weight and adipose tissue mass in obese mice [9,10]. This prompted us to assess whether differences in the reduction in adipose tissue volume during an exercise intervention could be related to differences in the exercise-induced production of Lac-Phe in humans, where a potential role of Lac-Phe has not been reported yet.

To this end, we quantified Lac-Phe after an acute bout of endurance exercise that was performed before and after an 8-week exercise intervention at 80% VO_2_peak in overweight subjects and subjects with obesity. Using this setup, we aimed to determine whether there are individual differences in the amount of Lac-Phe produced during physical exercise performed at the same relative cardiorespiratory intensity and whether these differences could, to some extent, explain the differences in adipose tissue reduction achieved during a lifestyle intervention in humans who are obese and/or overweight.

## 2. Materials and Methods

Study design and participants

The study was an 8-week supervised exercise intervention flanked by two acute exercise visits. Details of the study protocol have been published recently [11]. In brief, subjects had to be healthy, sedentary (<120 min habitual physical activity per week) and meet at least one of the risk factors: BMI >27 kg/m^2^, family history (first degree) of type 2 diabetes, and former gestational diabetes. Anthropometric parameters are presented in Table 1. Severe diseases were excluded through an anamnesis that included medication, routine laboratory parameters, electrocardiogram, and physical examination. Of the 26 subjects that met the criteria and completed the study, one subject was excluded due to newly diagnosed autoimmune thyroiditis and three due to incomplete blood sample sets, resulting in plasma sample sets from 22 subjects available for metabolomics analysis. All participants gave written informed consent and the study was approved by the ethics committee of the University of Tübingen and registered at Clinicaltrials.gov (NCT03151590).

Prior to and after the intervention, VO_2_peak was determined with metabolic gas analysis (MetaLyzer 3B and MetaMax 3B, Cortex Biophysics GmbH, Leipzig, Germany) during an incremental cycling test on an electromagnetically braked bicycle ergometer (Excalibur Sport, Lode BV, Groningen, Netherlands). VO_2_peak was defined as the mean VO_2_ over the last 20 s before the cessation of exercise due to volitional exhaustion or muscular fatigue. The intensity for training and acute exercise visits was individually set to 80% of the VO_2_peak determined at the initial performance test and controlled by the corresponding heart rate. The training intervention consisted of three times per week 1 h supervised endurance exercise, 30 min each cycling and walking, at 80% VO_2_peak.

The acute exercise visits were performed as follows: Blood was collected in the morning in the fasted state, 45 min before the commencement of exercise. In the meantime, the participants received a standardized breakfast (1 bun, 20 g butter, 1 slice of cheese, 150 g apple puree, water). Subsequently, they performed 30 min of bicycle ergometer exercise at 80% VO_2_peak and a second blood sample was collected 5 min after this bout of exercise. EDTA blood samples were immediately placed on ice, processed within 30 min and plasma stored at −80 °C.

Determination of insulin sensitivity using an oral glucose test and magnetic resonance imaging (MRI)-based assessment of whole-body adipose and lean tissue have been described [12]. MRI was performed using an axial T1-weighted fast spin-echo technique on a 3T whole-body imager (Magnetom Vida, Siemens Healthineers, Erlangen, Germany) with subjects in a prone position with extended arms. Segmentation of adipose and lean tissue was performed using an in-house developed procedure employing a modified fuzzy c-means algorithm [13] that classifies adipose and lean tissue of lower/upper extremities (feet to hips/shoulders to hands) and segments visceral and non-visceral (mainly consisting of subcutaneous fat) adipose tissue of the trunk in an automated fashion.

Quantification of plasma metabolites

The plasma levels of Lac-Phe, Lactate and Phe were determined using ultra high-performance liquid chromatography-mass spectrometry (UHPLC-MS)-based metabolomics [14]. A total of 50 µL of plasma were mixed with 250 µL of MeOH, vortexed for 30 s and centrifuged for 20 min at 16,000 g, 4 °C. The supernatant was vacuum-dried in aliquots of 200 µL. Dry samples were resuspended in 50 µL 25% ACN/water. The analysis was performed on a Vanquish UHPLC coupled to a Q Exactive (both Thermo Fisher Scientific, Waltham, MA, USA) operated in negative ion mode as previously described with slight modifications [14]. The separation was performed on a 2.1 × 100 mm ACQUITYTM UPLC HSS 1.8 µm T3 column (Waters, Milford, MA, USA). The mobile phases were (A) 6.5 mM ammonium bicarbonate in water and (B) 6.5 mM ammonium bicarbonate in 95% MeOH/water. The elution started with 2% B for 1 min, linearly changed to 100% B within 20 min, reverted back to 2% B, and equilibrated for 2.9 min (flow rate 0.35 mL/min, column temperature 50 °C). The Q Exactive was set to 140,000 resolution and full scan mode, mass scan range was 70–1050 m/z. Nitrogen sheath gas and nitrogen auxiliary gas were set at flow rates of 45 and 10 AU. Capillary and aux gas heater temperatures were 300 °C and 350 °C, respectively. The spray voltage was 3.00 kV. Parallel reaction monitoring was used to obtain high-resolution MS/MS spectra of Lac-Phe (m/z = 236.0928) with a resolution of 17,500 and a collision energy of 30 eV. The internal standard d5-Phe (615870, Merck, Darmstadt, Germany), 0.8 µg/mL in extraction solvent) was used to normalize signal intensities.

Statistical analysis

Statistical analyses were performed using JMP 16 (SAS Institute Inc., Cary, NC, USA). Longitudinal comparisons were performed using paired *t*-tests. Multiple linear regression analyses were performed on log-transformed data and adjusted for sex, age, baseline values of the respective tissue compartment or BMI, or change in muscle volume, as indicated. Normal distribution of the residuals was confirmed with the Shapiro–Wilk test in all analyses. A *p*-value < 0.05 was considered statistically significant.

## 3. Results

Lac-Phe was identified by LC-MS/MS (Figure 1A) and could be quantified in all samples (Figure 1B). Acute exercise caused a significant increase in plasma Lac-Phe levels, both before and after the 8-week training intervention (Figure 1B). The exercise intervention had no effect on Lac-Phe concentrations in the resting state or after the acute bout of exercise (Figure 1B), which was performed at the same relative intensity before and after the intervention. Plasma levels of Lac-Phe exhibited a significant correlation with phenylalanine (Figure 1C, R^2^ = 0.35) and a very strong correlation with lactate (Figure 1D, R^2^ = 0.82).

The training intervention resulted in an improvement in VO_2_peak and an increase in lean tissue, i.e., muscle mass, in the legs (Table 1). At the same time, BMI and abdominal subcutaneous and visceral adipose tissue decreased (Table 1). The decrease in subcutaneous adipose tissue during the intervention was inversely correlated with the concentration of Lac-Phe measured in blood plasma sampled immediately after acute exercise, both pre-training (Figure 1E) and post-training (Figure 1F) and also after adjustment for sex, age, and adipose tissue baseline values (results of multiple regression analyses shown in Table 2). The decrease in visceral adipose tissue was inversely correlated with the Lac-Phe concentration in plasma sampled after the post-training acute exercise bout and tended to be correlated after the pre-training acute exercise bout (Table 2).

Plasma lactate levels after acute exercise exhibited a similar but slightly weaker correlation with the change in abdominal adipose tissue during the 8-week intervention (Table 2). This association only reached statistical significance for the subcutaneous depot and only for lactate quantified in samples taken after the pre-training acute exercise bout but not for visceral adipose tissue or the post-training bout (Table 2). Furthermore, lactate levels after acute exercise were positively correlated with the increase in muscle mass of the lower extremities during the intervention. Lac-Phe levels after acute exercise still exhibited an inverse correlation with the change in subcutaneous adipose tissue when additionally adjusting for the change in leg muscle volume in the multiple linear regression models (*p* = 0.020, standardized Beta coefficient = −0.56 for the pre-training and a trend of *p* = 0.088, Beta = −0.42 for the post-training exercise bout).

No significant correlation of Lac-Phe or lactate could be observed with the changes in BMI or in the lean tissue of the arms (Table 2). As expected, the latter was not increased by the training scheme, i.e., by cycling and treadmill exercise (Table 1).

## 4. Discussion

Exercise-based lifestyle interventions aiming to reduce body weight in subjects with obesity exhibit a large variability regarding individual effectiveness, even when performed supervised and at the same relative cardiorespiratory intensity [2,3,4]. The metabolite Lac-Phe is produced during physical activity and has recently gained attention as a mediator of adipose tissue and weight loss in mice [9]. We hypothesized that the variability in the reduction of adipose tissue in humans with obesity participating in an exercise-only lifestyle intervention could be related to the amount of Lac-Phe acutely produced during exercise. We provide a first clue by showing that higher levels of Lac-Phe after acute exercise are related to a greater reduction in abdominal subcutaneous and, to a lower extent, visceral adipose tissue during an 8-week supervised training intervention.

Lac-Phe produced during physical exercise has been shown to reduce obesity by lowering food intake in mice [9], potentially via G protein-coupled receptors in the brain [10], and it is at least conceivable that higher levels of Lac-Phe did cause a greater transient suppression of hunger after each exercise session that contributed to a negative energy balance in our study. This transient repression of hunger, which serves the purpose of preserving blood flow to skeletal muscle, has been shown to correlate with the circulating concentration of lactate [15] and studies of lactate administration have supported such an appetite-suppressing effect [15,16]. Since lactate drives the formation of Lac-Phe, which then peaks after lactate [8,9], it could be speculated that Lac-Phe is a more sustained mediator of the appetite-suppressing lactate signal.

Since food intake was only assessed using questionnaires in our study, which could not be evaluated due to a high percentage of missing or implausible data, our results allow no conclusion regarding this potential mechanism. This is clearly a limitation of this work, as is the relatively small number of 22 subjects. Thus, future studies are required to substantiate our findings and elucidate the potential function of Lac-Phe as an “exerkine”.

Independent of its signaling function, Lac-Phe could serve as a biomarker to predict the individual response to exercise-based lifestyle interventions. This is particularly relevant given the large variability in the extent to which different subjects benefit from a given exercise scheme [2,3,4]. Exercise intensity is usually personalized, e.g., based on the individual VO_2_peak as in our study, but parameters to determine the ideal intensity or modality of exercise that are most suitable for an individual have not been available up to now. Lac-Phe could serve as such a parameter since subjects exhibiting higher levels after an exercise test achieved a greater reduction in adipose tissue mass despite having exercised at the same relative cardiorespiratory intensity. One aim of future studies could be to optimize the Lac-Phe response by modifying the exercise scheme or with the aid of dietary supplements to increase the effectiveness of exercise-based lifestyle interventions.

## Figures and Tables

**Figure 1 metabolites-13-00015-f001:**
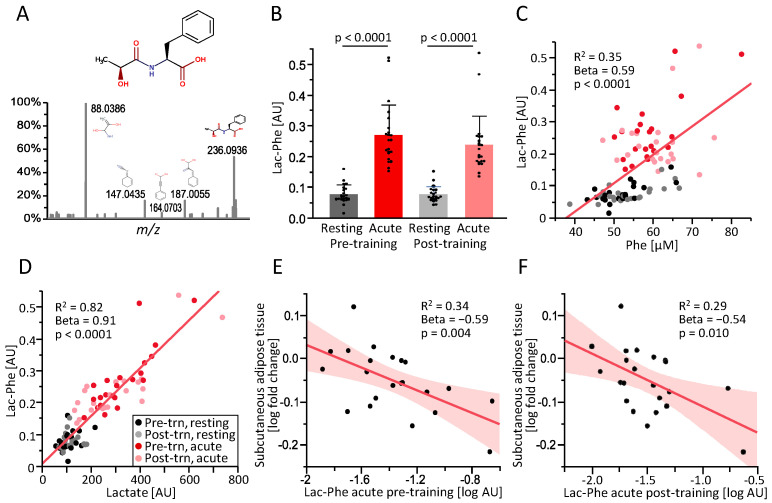
(**A**): Structure of N-Lactoylphenylalanine (Lac-Phe) and identification by LC-MS/MS. (**B**): Plasma levels of Lac-Phe in subjects with overweight and obesity (n = 22) before (resting) or immediately after (acute) a 30-min endurance exercise session before (pre) or after (post) an 8-week training intervention. (**C**,**D**): Correlation of plasma Lac-Phe with plasma phenylalanine (**C**) and lactate (**D**). (**E**,**F**): Correlation of the fold change in subcutaneous abdominal adipose tissue volume during the exercise intervention with the concentration of Lac-Phe after acute exercise before (pre-training, (**E**) and after the exercise intervention (post-training, (**F**). R^2^, standardized beta coefficient (Beta) and *p*-value are shown for the simple linear regression (values from multiple regression presented in Table 2). AU, arbitrary units.

**Table 1 metabolites-13-00015-t001:** Anthropometric, fitness and metabolic data.

	**Pre-Training**	**Post-Training**	***p*-Value**
Sex	14 female/8 male		-
Age [years]	30 ± 8.9 (19–59)		-
VO_2_peak/body mass [mL/(kg ∗ min)]	25.0 ± 4.2 (18.3–32.3)	26.5 ± 4.7 (16.0–34.9)	0.042 *
BMI [kg/m^2^]	31.7 ± 4.5 (27.5–45.5)	31.3 ± 4.7 (26.3–45.2)	0.006 *
Subcutaneous abdominal adipose tissue [L]	15.3 ± 5.9 (8.4–32.2)	14.7 ± 6.1 (7.2–33.1)	0.006 *
Visceral adipose tissue [L]	3.53 ± 1.65 (0.81–7.26)	3.38 ± 1.57 (0.94–6.68)	0.012 *
Lean tissue legs [L]	17.9 ± 4.1 (12.3–27.5)	18.2 ± 3.9 (13.2–27.7)	0.034 *
Lean tissue arms [L]	9.73 ± 1.96 (7.31–14.27)	9.85 ± 2.26 (7.19–14.42)	0.551
Glucose fasting [mmol/L]	5.09 ± 0.40 (4.61–6.00)	5.02 ± 0.40 (4.33–5.61)	0.336

N = 22, VO_2_peak N = 21; mean ± SD (range of values); * *p* < 0.05.

**Table 2 metabolites-13-00015-t002:** Association of the increased Lac-Phe and lactate levels after acute exercise with the training response.

Training Fold Change	Lac-Phe Pre-Training Acute	Lac-Phe Post-Training Acute	Lactate Pre-Training Acute	Lactate Post-Training Acute
Subcutaneous abdominal adipose tissue [L]	Beta = −0.62 *p* = 0.004 *	Beta = −0.52 *p* = 0.028 *	Beta = −0.60 *p* = 0.008 *	Beta = −0.39 *p* = 0.102
Visceral adipose tissue [L]	Beta = −0.42 *p* = 0.075	Beta = −0.48 *p* = 0.037 *	Beta = −−0.23 *p* = 0.372	Beta = −0.37 *p* = 0.123
BMI [kg/m^2^]	Beta = −0.25 *p* = 0.279	Beta = −0.15 *p* = 0.538	Beta = −0.13 *p* = 0.600	Beta = 0.07 *p* = 0.784
Lean tissue legs [L]	Beta = 0.37 *p* = 0.079	Beta = 0.22 *p* = 0.357	Beta = 0.42 *p* = 0.047*	Beta = 0.47 *p* = 0.036 *
Lean tissue arms [L]	Beta = 0.14 *p* = 0.584	Beta = 0.08 *p* = 0.758	Beta = 0.27 *p* = 0.279	Beta = 0.08 *p* = 0.759

Multiple linear regression analyses with adjustments for sex, age and baseline values of the respective tissue compartment or BMI. N = 22; Beta, standardized beta coefficient; * *p* < 0.05.

## Data Availability

The data will only be made available to interested researchers upon reasonable request as far as privacy and consent of research participants are not compromised. The data presented in this study are not publicly available due to them containing information that could compromise privacy and consent of research participants.

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
