# Peer review of "Exercise-Induced N-Lactoylphenylalanine Predicts Adipose Tissue Loss during Endurance Training in Overweight and Obese Humans"

_metabolites, 2022, doi:10.3390/metabo13010015_

Round 1
Reviewer 1 Report
This is a novel topic, that could be of relevance with the ever increasing prevalence of obesity.
The introduction is clear and easy to follow. I think that you have too many key words and would be best to stick to no more than 5 of these.
In line 42 you suggest that there may be adverse reactions from increasing the exercise dose. I am not sure what they could be and think they need to be stated.
You have stated that the study protocol has been already published, but this is not another of your investigations but a different team. I feel it would be better to state the inclusion and exclusion criteria, as well as how VO2 peak was attained and also a clearer outline of the 8 week protocol. The reader does not wish to access another paper to find out this information which is pivitol to your study.
In the methods you have stated that adipose tissue was segmented by an automated procedure and then referenced (11), it would be good to at least give the name of the procedure.
Throught the paper you have referred to untrained and trained so that when I came to look at the results I was expecting two groups. However, I realise that it is all one group and what you mean it pre and post training intervention. I believe that this terminology should be used to inform the reader.
In the results you state that training had no effect on Lac-Phe in both resting or after acute training both pre or post intervention. If this is the case your discussion does not agree with this as you have said that those with higher Lac-Phe after exercise had greater adipose tissue reduction, yet you have said above that there is no change with exercise. This needs clarifying in the results and the discussion.
Line 147 - why would you expect to see a change in abdominal tissue with lactate levels?The associaiton with statistical significance for subcutaneous deposits after the pre-training acute exercise bout doesn't really mean anything as the reduction in abdominal tissue would not have changed at pre-training and you have said that there was no change in Lac-Phe with training. The association with lactate pre and post training for the acute session with lean tissue in the legs and arms does not really show anything. It is a normal process to produce lactate when exercising and the change in lean tissue would be due to the chronic training programme that had been undertaken.
Line 133- You have stated that the partcipant lost subcutaneous adipose tissue during the 8 week training intervention and that you found a correlation with Lac-Phe, however as lactate is produced as a by-product of exercise, I do not feel that this finding actually shows anything.
In the discussion you talk extensively about food intake, yet you have not looked at this and some of your claims around appetite changes are not relevant to this group of people. I believe that this should be omitted as there are too many assumptions being made and it is not an a priori to your research question. This may make an interesting future investigation.
I am sure that the above will make the outcome from your investigation clearer
Author Response
First of all, we would like to thank this reviewer for finding merit in our work and providing helpful comments. We address these in our point-by-point response below.
This is a novel topic, that could be of relevance with the ever increasing prevalence of obesity.
The introduction is clear and easy to follow. I think that you have too many key words and would be best to stick to no more than 5 of these.
- Keywords have been shortened.
In line 42 you suggest that there may be adverse reactions from increasing the exercise dose. I am not sure what they could be and think they need to be stated.
- Thank you for this comment. The cited reference uses the term “adverse” to denote a worsening in parameters such as glucose control. We were also thinking of overtraining or cardiac events, which are -hopefully- unlikely to occur in supervised lifestyle interventions. We therefore decided to omit this aspect and changed the wording accordingly (lines 37-37) .
You have stated that the study protocol has been already published, but this is not another of your investigations but a different team. I feel it would be better to state the inclusion and exclusion criteria, as well as how VO2 peak was attained and also a clearer outline of the 8 week protocol. The reader does not wish to access another paper to find out this information which is pivitol to your study.
- Done (lines 77-113).
In the methods you have stated that adipose tissue was segmented by an automated procedure and then referenced (11), it would be good to at least give the name of the procedure.
- The method is an in-house developed, fully automated and standardized procedure based on a modified fuzzy c-means algorithm. This information has been added to the new version of the manuscript (lines 105-113).
Throught the paper you have referred to untrained and trained so that when I came to look at the results I was expecting two groups. However, I realise that it is all one group and what you mean it pre and post training intervention. I believe that this terminology should be used to inform the reader.
- Terminology has consistently been changed to “pre/post-training” (Figure, tables and text).
In the results you state that training had no effect on Lac-Phe in both resting or after acute training both pre or post intervention. If this is the case your discussion does not agree with this as you have said that those with higher Lac-Phe after exercise had greater adipose tissue reduction, yet you have said above that there is no change with exercise. This needs clarifying in the results and the discussion.
- Lac-Phe was, indeed, similar when comparing resting pre- and post-training and acutely exercised state pre- and post-training , respectively. However, Lac-Phe that was produced during and accumulated after acute exercise was negatively associated with the change in adipose tissue volume during the intervention, i.e., a higher amount of acute exercise-induced Lac-Phe predicted a greater adipose tissue loss during the 8-week exercise intervention. We have attempted to make this more clear in the results (lines 144, 156, 164-170, 176-179) and discussion (lines 203-205).
Line 147 - why would you expect to see a change in abdominal tissue with lactate levels?The associaiton with statistical significance for subcutaneous deposits after the pre-training acute exercise bout doesn't really mean anything as the reduction in abdominal tissue would not have changed at pre-training and you have said that there was no change in Lac-Phe with training. The association with lactate pre and post training for the acute session with lean tissue in the legs and arms does not really show anything. It is a normal process to produce lactate when exercising and the change in lean tissue would be due to the chronic training programme that had been undertaken.
Line 133- You have stated that the partcipant lost subcutaneous adipose tissue during the 8 week training intervention and that you found a correlation with Lac-Phe, however as lactate is produced as a by-product of exercise, I do not feel that this finding actually shows anything.
- Thank you for this remark. Indeed, the only novel factor in our study is Lac-Phe. However, since Lac-Phe is produced from lactate and phenylalanine and since Lac-Phe concentrations were strongly correlated with lactate in our study (Fig. 1D), we believe it is relevant to also assess the association with adipose tissue reduction for lactate. As for Lac-Phe, we tried to make the distinction between acute exercise- and training-induced changes more clear (lines 176-179).
In the discussion you talk extensively about food intake, yet you have not looked at this and some of your claims around appetite changes are not relevant to this group of people. I believe that this should be omitted as there are too many assumptions being made and it is not an a priori to your research question. This may make an interesting future investigation.
- While we find this aspect quite exciting and whilie it would be in accordance with the data published in rodents (Ref. 9 in the manuscript) we agree that this part of the discussion is highly speculative. We did, therefore, cut it down and underline its speculative character. In fact, the subjects were asked to fill in food questionnaires at the beginning and end of the study but these were of poor quality. In the hindsight, a more sophisticated technique would have been required to systematically and reliably assess food intake. This is now also discussed as a limitation (216-218).
I am sure that the above will make the outcome from your investigation clearer
Thank you.
Reviewer 2 Report
The study of Hoene et al aims to correlate th post-exercise (acute) variations of specific metabolites (lac-phe) with changes in adipose tissue mass. The study is quite novel and potentially interesting.
However, there are a number of critical aspects to address:
1. Introduction does not provide sufficient background evidence to make readers familiar with the argument and properly support the need for such intervention.
2. The authors should clearly state the aim of their intervention.
3. Materials and methods are lacking information: the authors should report the characteristics of the participants, the inclusion/exclusion criteria etc.
4. The design of the study, timing of intervention and measurement result unclear from the description, and this affects the understanding of following sections.
5. How was the basal level of physical activity assessed? Please address this aspect.
6. What about changes in the dietary habits? How did you control this aspect?
7. How did you assess the “target heartrate for each subject” and which instruments have been used to assess all the measurements?
8. The exiguous number of subjects (22) underpowers the statistic and does not allow tracing a strong correlation. This aspect is critical when tracing correlation of transient changes of metabolites with stable changes (adipose tissue mass). This is importantly related to the lack of size sample determination analysis. Please address this aspect.
9. Limitations of the study should be listed in the discussion section.
I offered these critiques in a constructive spirit, hoping they might be of help for the Authors.
Author Response
First of all, we would like to thank this reviewer for finding merit in our work and providing helpful comments. We address these in our point-by-point response below.
The study of Hoene et al aims to correlate th post-exercise (acute) variations of specific metabolites (lac-phe) with changes in adipose tissue mass. The study is quite novel and potentially interesting.
However, there are a number of critical aspects to address:
- Introduction does not provide sufficient background evidence to make readers familiar with the argument and properly support the need for such intervention.
- Thank you for having drawn our attention to this issue. We have revised the introduction and hope that both background and aims (comment 2) are now more clear (lines 40-74).
- The authors should clearly state the aim of their intervention.
- Please see response to 1.
- Materials and methods are lacking information: the authors should report the characteristics of the participants, the inclusion/exclusion criteria etc.
- Done.
- The design of the study, timing of intervention and measurement result unclear from the description, and this affects the understanding of following sections.
- We have attempted to make the study design more clear in the results section and also by stating the design more explicitly in the abstract (line 24), introduction (lines 68-70) and discussion (lines 203-205), knowing that the methods part is not always read in detail.
- How was the basal level of physical activity assessed? Please address this aspect.
- Basal level of structured physical activity was assessed using a questionnaires and activity <120 minutes/week employed as an inclusion criterion. In addition, the basal VO2 peak of 25.0±4.2 (range 18.3 – 32.3) mL/kg*min (Table 1) is a clear indicator of a low level of physical activity. Subjects were instructed not to alter their basal physical activity habits during the intervention.
- What about changes in the dietary habits? How did you control this aspect?
- The participants were instructed not to alter their eating habits during the intervention. As a support, they were asked to fill in food questionnaires before and after the intervention. Unfortunately, these self-reports were of pure quality and not suitable for statistical evaluation, which is also discussed in the new limitations section (lines 216-220).
- How did you assess the “target heartrate for each subject” and which instruments have been used to assess all the measurements?
- Prior to the intervention (i.e. prior to the first acute exercise visit), VO2peak was determined by metabolic gas analysis during an incremental cycling test on an electromagnetically braked bicycle ergometer. The test was terminated due to volitional exhaustion or muscular fatigue. VO2peak was defined as the mean VO2 over the last 20 s before the cessation of exercise. The heart rate corresponding to 80% of VO2peak during this initial performance test was employed as the individual target rate. Heart rate was continuously monitored during all exercise sessions and bicycle “speed” was continuously adjusted to maintain this predetermined heart rate. This, as well as the instruments used, is now also detailed in the revised methods (lines 89-97).
- The exiguous number of subjects (22) underpowers the statistic and does not allow tracing a strong correlation. This aspect is critical when tracing correlation of transient changes of metabolites with stable changes (adipose tissue mass). This is importantly related to the lack of size sample determination analysis. Please address this aspect.
- We agree that this is a limitation of the study and hope that our report will stimulate further studies on Lac-Phe. We have now added a limitations section to the discussion where this point is addressed (lines 216-220).
- Limitations of the study should be listed in the discussion section.
- Done, lines lines 216-220 in the new version.
I offered these critiques in a constructive spirit, hoping they might be of help for the Authors.
Thank you.
Round 2
Reviewer 1 Report
I am pleased to see the changes that have been made and feel it is much improved and easier to understand.
Author Response
Thank you very much.
Reviewer 2 Report
I acknowledge the substantial efforts made by the Authors to improve their manuscript. I reckon it may be now considered for suitable for publication , providing some minor issues will be addressed.
1) The theoretical framework in the introduction keeps suffering from outlining key beneficial effects exerted by exercise.
2) In the methods section: which questionnaire was used to assess physical activity levels? Please reference it.
Author Response
I acknowledge the substantial efforts made by the Authors to improve their manuscript. I reckon it may be now considered for suitable for publication , providing some minor issues will be addressed.
-> Thank you very much.
1) The theoretical framework in the introduction keeps suffering from outlining key beneficial effects exerted by exercise.
-> We have added the following to the introduction (lines 2-3): ]. “In addition to increasing energy expenditure and reducing adipose tissue volume, physical exercise exerts a multitude of benefical effects on the whole-body level and counteracts type 2 diabetes by improving insulin resistance and blood glucose control.”
2) In the methods section: which questionnaire was used to assess physical activity levels? Please reference it.
-> Habitual physical activity was assessed employing a German -language questionnaire based on (J A Baecke, J Burema, J E Frijters. A short questionnaire for the measurement of habitual physical activity in epidemiological studies. Am J Clin Nutr. 1982 Nov;36(5):936-42. doi: 10.1093/ajcn/36.5.936.)